# How to handle uncertainties in modelling due to human reliability issues for nuclear disposals

Oliver Straeter[1], Fabian Fritsch[1]

[1] Work- and organizational Psychology, University Kassel, D-34132 Kassel, Germany

*Correspondence to*: Oliver Straeter (straeter@uni-kassel.de)

**Abstract.** Modelling plays a crucial role in assessing the design for future technical or geological development of a repository for radioactive waste. Models and the application of these models to scenarios are used to weigh different safety-related designs or to assess the suitability of sites. Even if the models simulate and evaluate a system in a geological context that is designed as a passively safe system, the human factor plays a significant role in the overall assessment process and thus in finding a site

with the best possible safety in accordance with the Site Selection Act. This influence is not seen at the level of repository operation, as is traditionally viewed with regard to human factors, but rather in the design of the repository - particularly in the decision-making process and the definition of the system's fundamental design parameters. Thus, considerations of human reliability are also of utmost importance for the passively safe system of a repository, especially in the current phase of the search and evaluation process. Given that severe accidents in man-made technological systems depend heavily on the reliability

of human behaviour, not only in operation but also in design and conceptualization, considering human reliability aspects is essential for a successful site selection (Straeter, 2019).

This article first provides an overview of the technical, organizational, cross-organizational, and individual aspects of human reliability that are crucial in the modelling phase of radioactive waste management. Human aspects include variations in the selection of models, the definition of input parameters, and the interpretation of results as individual or group efforts. Based

on a review of relevant guidelines on the topic (VDI 4006), suggestions are presented for dealing with these human factors at different levels. The approach is supported by a study on the importance of these factors, which was carried out in the context of the TRANSENS project in cooperation of the University of Kassel and the TU Clausthal. Overall, based on these considerations, the AHRIC (Assessment of Human Reliability in Concept phases) method is proposed to assess the negative effects of trust issues in the site selection work processes and to derive mitigating measures (Fritsch, 2025). The method applies

to all work processes of the key actors.

## 1 Introduction

The search process for high-level radioactive waste in Germany is characterized by the selection and interpretation of criteria, geological assessments, data interpretation, and modelling of the effects of geological and technical barriers. Even if an

operational mine or the disposal process is not yet underway, this process is nevertheless characterized by diverse human influences on the reliability of the sub-process. Examples of problems include:

- Decision-making processes and preferences, as well as assessment of limitations of predictive models
- Uncertainties in the use of predictive models used to define design criteria
- Decision-making in reconciling scientific results with other (political or societal) constraints
- Conceptualization and design of disposal processes and facilities in a materials science and geological context
- Prospective assessment of the impacts of operational aspects of storage and disposal

Operational procedures for assessing human reliability require defined operational conditions (contexts) to estimate the safety impact. These conditions are evaluated based on the specification of activities (work-as-planned) and the implementation of activities in the real environment (work-as-done). VDI 4006 (2025) defines work-as-planned as task-oriented behaviour and work-as-done as goal-oriented behaviour. In the latter case, in addition to the specified activity, requirements from the real environment lead to the need to balance different goals in the work context.

During the search process, the emplacement processes are not yet as precisely specified as would be necessary for an assessment using operational procedures. Nevertheless, the influences of this search and planning phase are crucial and determine essential safety-related properties of the repository.

An assessment of human reliability during the site selection phase is therefore essential. The following section first addresses the issues of the modelling process and the importance of trust. It then shows how an assessment of human reliability can be achieved in this planning phase of the search for a final repository.

## 2 Human Reliability Issues in a Modelling Process

### 2.1 Issues of a Modelling Process with importance to Human Reliability

Human reliability issues in modelling are naturally different from those encountered in the operational management of tasks in a real mine or packaging plant. In order to structure these issues and demonstrate their significance, the steps of a typical modelling process can be used as a basis. When modelling - for example, geological facts or the design of a container - different steps must be completed, starting with the scope of consideration and ending with the interpretation of the modelling results. Different people with different concepts and previous experience are involved in these processes.

These individuals must interpret and evaluate the uncertainties of the modelling using their cognitive concepts and prior individual experiences. In modelling, the completeness and assumptions about system behaviour always play a decisive role, as does the fit of these concepts and experience for assessing the accuracy of a model (Straeter, 2005). Table 1 shows the human reliability issues that can be expected in modelling.

As the table illustrates, different weighing processes must be performed under uncertainty within the framework of a variety of modelling approaches. The people involved in this modelling process must carry out these weighing processes on the basis

of their expertise. Their own expertise, consisting of experiences and the concepts developed from them, plays a crucial role, because people are bound to the cognitive processing cycle within the framework of their cognitive processing (Straeter, 2005). This states that an assessment is always based on concepts acquired by themselves, and any mismatch between these and the information from the world is compared via a central comparator (the limbic system). The comparison always occurs both cognitively and emotionally. The emotional component means that mismatches are evaluated not only cognitively but also emotionally. This results in the devaluation of inappropriate information (under-trust), the search for confirmation of one's own experiences and concepts (over-trust), and the search for simplifications to reconcile one's own experiences and concepts with information. The way in which these simplification strategies take place is known in psychology as biases (Tversky & Kahneman, 1974; Englisch, 2024). Biases enable us to efficiently deal with mismatches in our environment based on our 'gut feeling'. On the other hand, this effect pretends us a degree of confidence based on our own experiences and concepts in a given situation that is not necessarily valid.

**Table 1: Expected issues regarding human reliability in modelling.**

| Modelling steps | Issues with impact on Human Reliability |
|---|---|
| ▪ Scoping of issue (purpose of model) | ➢ completeness & assumptions |
| ▪ Choice of (set of) model(s) for issue | ➢ fit / validity |
| ▪ Computerized realization | ➢ correctness of code |
| ▪ Parametrization of Model(s) | ➢ correctness of settings |
| ▪ Calculation depth (iterations / variations) / sensitivity analysis | ➢ sufficient time (empowerment) & sufficient variations |
| ▪ Result generation | ➢ precision of calculation |
| ▪ Results interpretation | ➢ sufficient experience |
| ▪ Decision on reevaluation / further evaluation / further depth | ➢ sufficient criteria, feedback & time |

**2.2 Trust and Human Reliability**

Correct dealing with this mismatch is an issue of correct trust into oneself and the information provided to us in a particular given situation. The highest level of reliability is achieved by building a reasonable relationship of trust. Over- and under- trust results into biases. For the task of modelling, this means, for example:

- Under-trust results into scenarios where appropriate models are not used
- Over-trust result into scenarios where inappropriate models are used

Figure 1 according to EUROCONTROL (2003) describes this relationship between the fit of one's own abilities (experiences and concepts) on the x-axis and the cognitive-emotional regulation (dashed solid ogives) and the resulting behaviour on the y-axis.

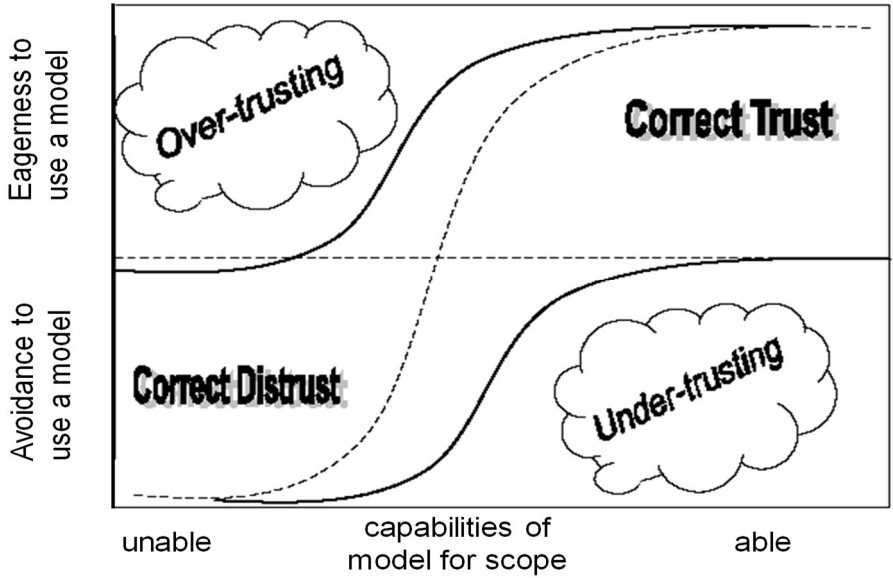

Figure 1: Relationship between own abilities (experiences and concepts) on the x-axis and the trust related cognitive-emotional regulation (dashed solid ogives) and the resulting behaviour on the y-axis (based on EUROCONTROL, 2003).

The human cognitive processing cycle performs a comparison and assessment once a mismatch exists. But there is a biased relationship to the behaviour of a person depending on the level of trust. Reliability issues (VDI 4006) follow from inappropriate trust alignment (i.e., over-trust or under-trust) may result into avoidance (under-trust) or overconfidence (over-trust). The behavioural regulation is not a rational, task-oriented one; rather more, the regulation is goal-oriented: in over-trust behaviour, one is too over-confident with one's own competencies; in under-trust situation one is diminishing even valid alternatives. Resulting behaviour in modelling may be, but are not limited to:

- Promoting abilities of own model and defending model disabilities in over-trust scenarios
- Seemingly reliable modelling results if over-trust into own models
- Arguments speaking for other models neglected if over-trust into own models
- Diminishing other models in under-trust scenarios
- Diminishing people representing other models in under-trust scenarios
- Seeking for confirming arguments for own model in over-trust and under-trust scenarios

Factors determining Trust Issues can be distinguished into individual, team or leadership issues. The safety-critical impacts of such behaviour in the context of the site selection process for highly radioactive waste are:

- Unfavourable selection of models for the assessment of the site selection
105 - Inappropriate use of model
- Lack of reflection on assumptions of the models
- No consideration of interactions between different parameter
- Interpreting the results in a favourable manner
- Finger pointing on other models or diminishing other opinions
110 - Exclusion of other models

On the individual perspective the cognitive processing cycle lets conclude that own experiences and concepts are of importance (Straeter, 2005). Given this, Picture 2 shows salient individual factors determining trust based on a study on (EUROCONTROL, 2003). Self-confidence on own experience and concepts may lead to over-trust if own is too self-confident;
115 on the other hand - if too less confident of own capabilities, one might to under-trust oneself. Second parameter is the understanding of the situation. Finally, the capabilities of the model (respectively the competence of the tool) may determine over- as well as under-trust in own capabilities. The same logic holds for capabilities of others.

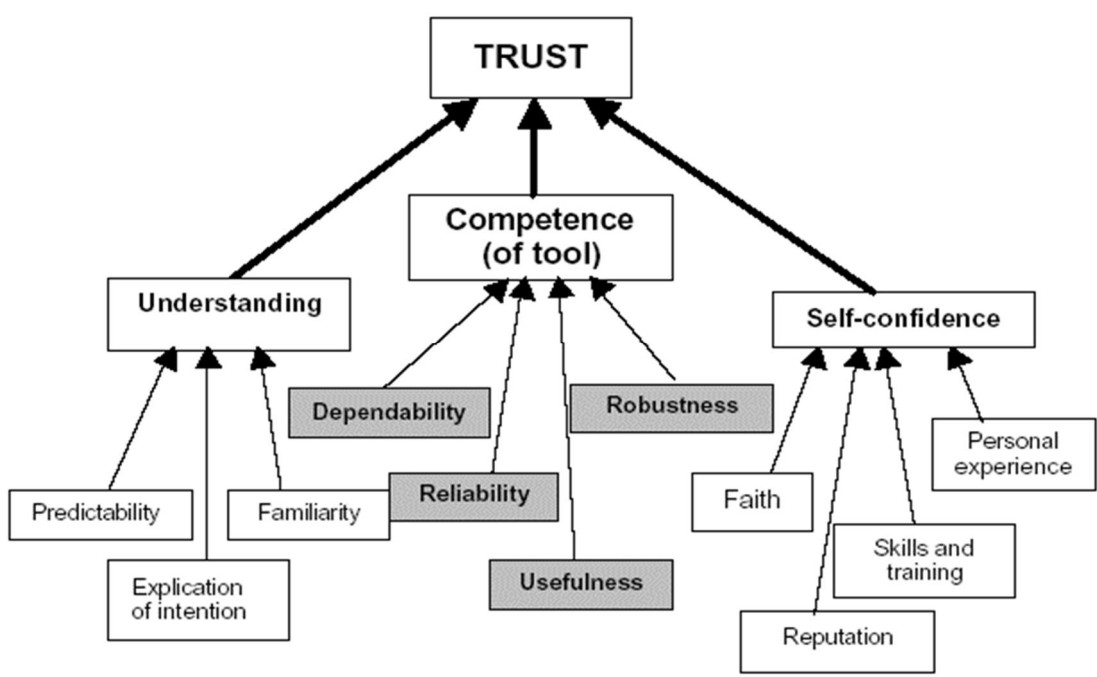

120 **Figure 2: Individual factors determining trust (EUROCONTROL, 2003).**

Group perspective has, in addition to the individual factors, a couple of psychological dimensions triggering trust. Jäckel (2016) investigated this with focus on team dynamics and leadership issues, whereas leadership in this study was understood as the lead in a team-dynamics by a certain person (hence not necessarily the hierarchical leadership). She evaluated that trust depends on a couple of factors, shown in Figure 3.

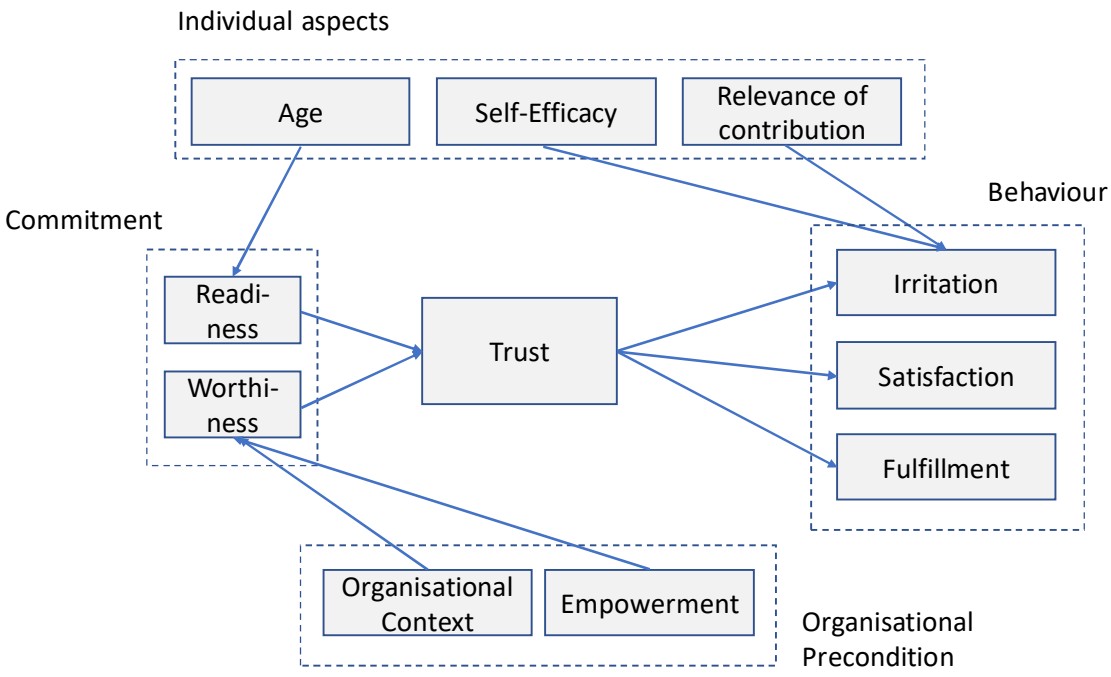

**Figure 3: Group leadership factors determining trust in a leading opinion or person (adapted from Jäckel, 2016).**

## 2.3 Impact of trust on the reliability of the modelling process

A case study conducted together with the Department of Geomechanics and Multiphysical Systems of TU-Clausthal showed the safety criticality of trust for the result in modelling a critical parameter for the design of the vessel for a final disposal of radioactive waste (Muxlhanga et al, 2024).

As part of the TRANSENS project, numerical simulations of a disposal roadway for a radioactive waste repository demonstrated that there is a significant individual influence on the simulation results, even if the same technical conditions (material data, material models, simulator) are applied. The experiments revealed an uncertainty factor of 1.2 to 2.5 for the target variable of vertical convergence of the roadway crests. This uncertainty was the result of a simple determination of an amplification factor (slope of a straight-line equation) based on a series of approximately 14 data points. The results are highly

relevant for modeling creep processes and their impact on the computationally determined load-bearing behavior of a roadway in the Salinar Mountains.

In line with the aforementioned modeling phases (see above Figure 1), the parameterization of the model was addressed in this study. The general conclusion from this study is that influences based on individual decisions and approaches of the respective modeler should be considered in the modeling, as these have safety-relevant effects on the design of a repository.

Figure 4 shows how this can be implemented in an assessment. The various modeling phases can be viewed as system functions summarized in a flowchart. Flowcharts are commonly used to describe safety-related impacts in complex systems (e.g., GRS, 1989). System functions can function (correct trust scenarios) or fail (over- or under-trust scenarios). Overall, the general modeling flowchart results in various safety-relevant end states of the model.

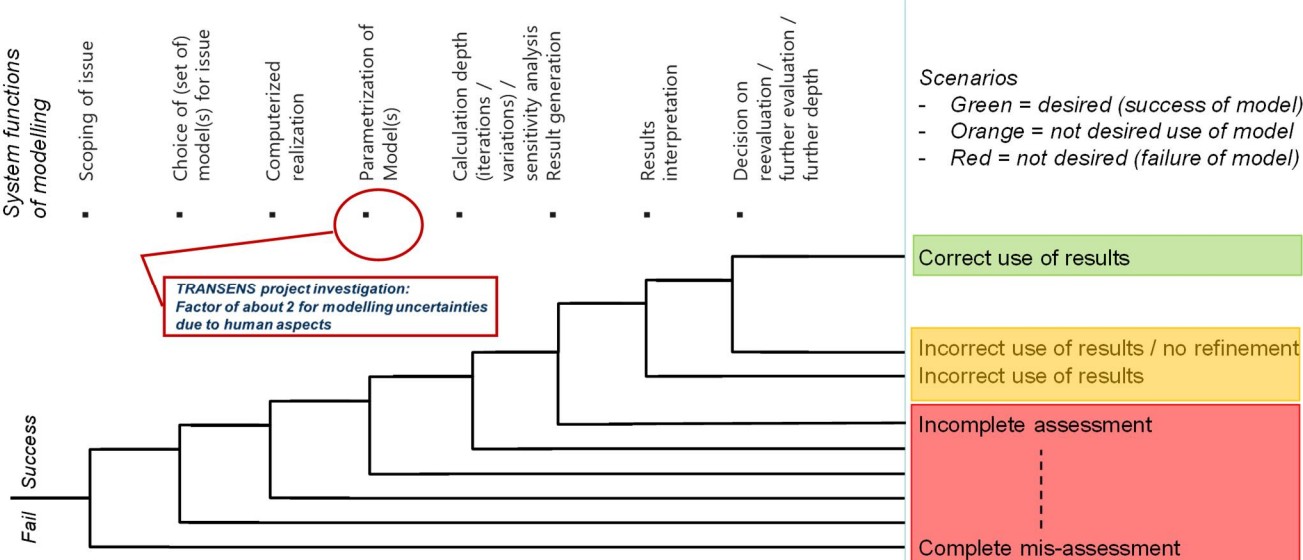

**Figure 4: Overall approach to assess trust and biases in modelling using an event-sequence diagram.**

Trust issues lead to biases that might impact the reliability of a modelling process and hence determines the outcome of the judgement of best possible safety of a final disposal site. A systematic assessment and mitigation of such biases is hence utmost required. The following chapter outlines a method how to deal with trust issues and biases in the current phase of the site selection.

## 3 Systematic Assessment and Mitigation of Biases using AHRIC

### 3.1 Assessment of biases

The AHRIC (Assessment of Human Reliability in Concept phases) method was developed to identify uncertainties due to human factors and to evaluate the described biases in context of geological modelling. The starting point was the questions

regarding uncertainties within the framework of the URS research cluster (Uncertainties and Robustness with regard to the Safety of a repository for high-level radioactive waste - https://urs.ifgt.tu-freiberg.de/en/home; Kurgyis et al., 2024). The issues relating to this cluster and the site selection in general present the following challenges, among others:

- no existing system that could be assessed at present or in the coming decades
- complex model calculations for long-term analyses based on parameters with a wide range (e.g., rock parameters)
- dependencies and interactions between the different barriers for the safe containment of waste (e.g., containers and host rock as barriers).

AHRIC enables the assessment and increase of human reliability throughout the modelling phases described in Figure 1. This
provides a comprehensive view of human reliability that previous methods have not covered. The AHRIC method is a self-assessment questionnaire designed for all modelling activities in the site selection process. It is a universally applicable framework that can be applied to a variety of system developments and industries. Therefore, in addition to geological models, it is also possible to evaluate all other conceivable modelling activities within the context of nuclear waste disposal.

The contents have been compiled from the literature on human information processing in such modelling processes and
following classical steps of questionnaire development (Bühner, 2021). The AHRIC is divided into three primary categories, each of which has its own set of aspects (biases). Table 2 provides an overview of the AHRIC and presents an excerpt of its contents. Additionally, the significance of the bias or aspect is described in simplified terms, with each being predominantly measured using two items.

A well-known example of the first main category is the confirmation bias. It describes the unconscious human tendency to
search for information one-sidedly and to interpret it in such a way that an assumption or hypothesis once made is confirmed (Evans, 1990). This process unfolds through three key tendencies (Hager & Weißmann, 1991):

- Preferential selection of confirmatory information (e.g., targeted search for data that fit the model assumption)
- Enhancing the value of confirmatory and matching information (e.g., studies that show matching data are given greater value)
- Interpretation of contradictory information as confirmatory information (e.g., reinterpretation of unexpected model results as outliers that confirm the rule).

In the second main category, the phenomenon of groupthink is an important component. The desire for uniformity in cohesive groups becomes so strong that realistic decisions are made more difficult and decision alternatives are disregarded (Janis,
1971). Groupthink is a phenomenon that occurs irrespective of the abilities of the group members, such as intelligence or experience. It can lead to management errors, even in large corporations (Peterson et al., 1998). The result is an unreliable decision-making process or a decision of poor quality.

**Table 2: Overview of AHRIC (excerpt).**

| Category | Bias/aspect | Description |
|---|---|---|
| Individual level | Anchoring bias | Distorts judgement by over-relying on initial information |
| | Confirmation bias | Favouring information that confirms existing hypotheses or assumptions |
| | Escalating commitment | Persisting in a failing course of action due to prior investments |
| Group level | Groupthink | Undermines critical thinking in order to maintain group harmony |
| | Group polarisation | Group format leads to more extreme decisions (overestimation or underestimation) |
| | Obedience to authority | Complying with directives from authority figures despite personal doubts |
| Situational level | Management objectives | Decisions are skewed by the pressure to meet management objectives |
| | Ad hoc dynamics | Influence through spontaneous and inappropriate solutions |
| | Time pressure | Reduced quality of decision-making due to time pressure |


In the third main category, management objectives that must be achieved despite internal contradictions serve as an example. In addition, time pressure plays a role as a classic trigger for goal-oriented behaviour. These aspects must be additionally accounted for in the context of trust in models, as they contribute to an increased probability of biases emergence.

In principle, all aspects of the AHRIC method are formulated for self-assessment, so that the statements are answered by the persons concerned themselves. This makes it possible to evaluate one's own modelling. On the one hand, this enhances confidence in one's own modelling; on the other hand, it can strengthen the trust of other scientists or the general public in the modeller. There is a 6-point Likert scale available for responding to self-statements. These statements are rated in terms of the level of agreement with them. Disagreement with an item (e.g., "I have made specific considerations that contradict the initial

information.") is interpreted as an indication of the corresponding bias, in this case the anchoring bias. Figure 5 shows the systematics of the AHRIC.

This classification system has been developed for the purpose of achieving measurement sensitivity, thereby facilitating precise measurement. Simultaneously, the system has been designed to be straightforward and uncomplicated. This is important for the interpretation of the results. In the web-based version of AHRIC, the interpretation is carried out automatically by a back-

end code, and in a paper-based survey, the users of the method can easily carry out the interpretation themselves.

| 1 | 2 | 3 | 4 | 5 | 6 |
|---|---|---|---|---|---|
| completely agree | mostly agree | agree to some extent | disagree to some extent | mostly disagree | completely disagree |
| No indication of susceptibility to the respective bias | | | Slight indicator | Strong indication of susceptibility to the respective bias | |

**Figure 5: AHRIC response scale with data interpretation.**

In order to address the factors that determine trust, as described in Chapter 2.2, the AHRIC method is available in two versions:

- AHRIC-M (Medium Version) takes 8-12 minutes to complete
- AHRIC-S (Short Version) which takes approx. 2 minutes to complete

As a self-assessment tool, AHRIC-M contains all the essential components to assess one's own reliability in the context of
modelling. The AHRIC-S is available for teams, groups or departments. This version focuses primarily on group dynamics aspects as a group resilience monitoring. This makes it possible to evaluate group meetings or decisions quickly, for example at all modelling milestones.

### 3.2 Procedure of an application in modelling
AHRIC provides an assessment framework for each modelling stage. An example how it is able to assess an overall project-management process is demonstrated in Figure 6. For the success probabilities shown in Figure 6, hypothetical values are assumed. A success probability of $p = 0,857$ indicates that six out of seven biases are unremarkable, while one bias dominates in the corresponding modelling phase. In the second modelling step, $p = 1$ is assumed as the success probability. The right-hand column shows the outcomes derived from multiplying the probabilities.

There are various ways to manage and execute modelling, such as using classical or agile methods. Regardless of the specific approach, both AHRIC-M and AHRIC-S can be used to assess human reliability in a goal-oriented manner.

AHRIC-M is designed primarily for individuals to evaluate their own activities at reasonable intervals, such as every six months, or at appropriate milestones or modelling stages (i.e. decision points).

AHRIC-S, the short version, offers a valuable opportunity to assess group reliability and contribute to group resilience and
trust throughout all steps involving collective discussion and decision-making.

Classically, such group processes take place very often in the context of developments, e.g., in the form of weekly meetings. Ongoing evaluation within the group can also lead to a progression of data from which it can be seen, for example, that group members are thinking and judging in an increasingly uniform and adapted way. In terms of group polarisation, this usually leads to more extreme and risky decisions (Jonas et al., 2014).


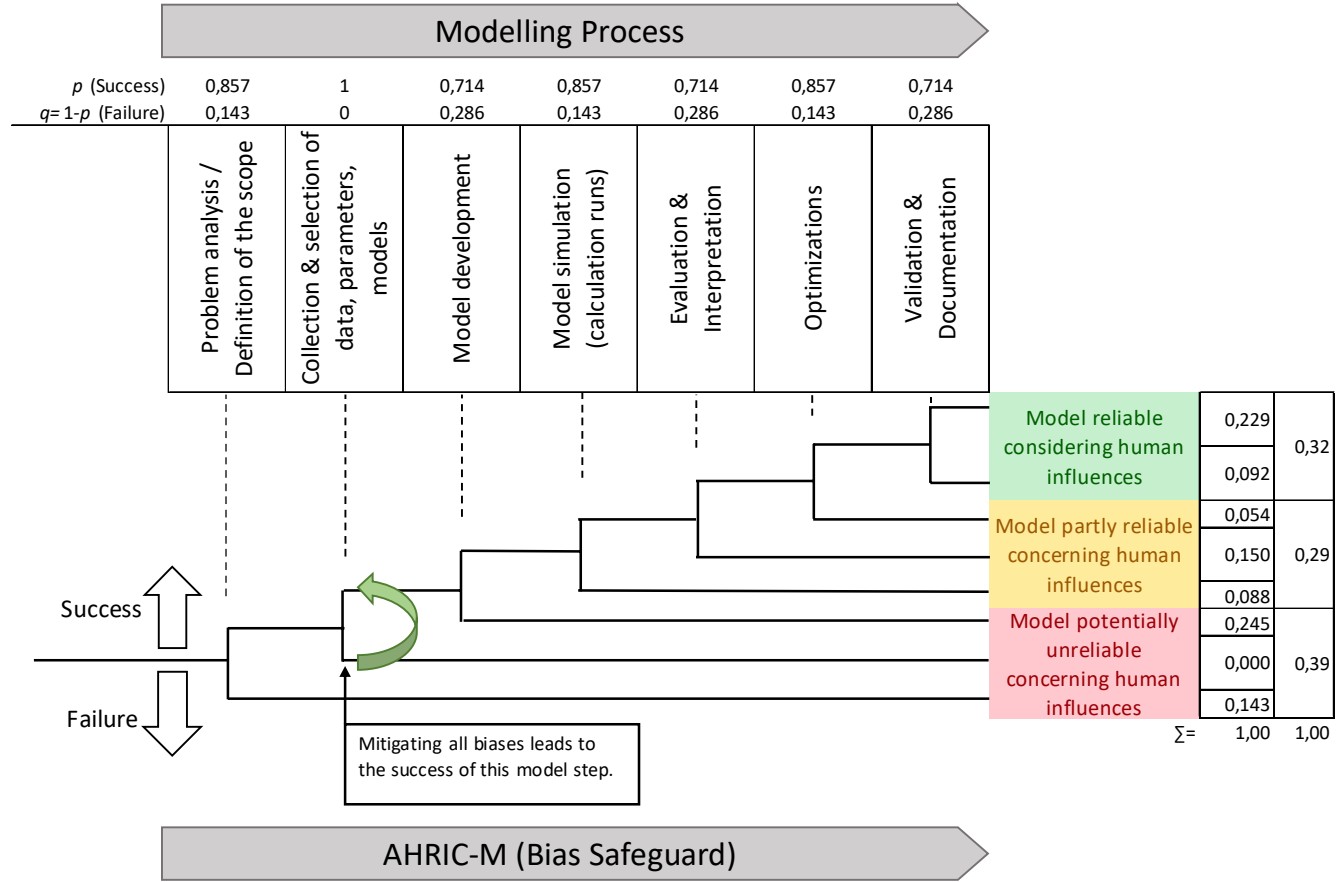

**Figure 6: Illustrative example of an AHRIC implementation within the modelling process.**

Looking at the contribution made by AHRIC to the modelling process, it is possible to evaluate the success or failure of models.
This relates to human influence and results from the number of biases present in each step of the modelling process. An example of such a representation, using hypothetical data to illustrate the individual steps of the modelling process, showed the relevance and fit to observable date in project-management (Dierig, 2014). Mitigations can be achieved in two ways using AHRIC:

- Using AHRIC-M as bias safeguard if measurements results in no show of bias or
- Bias mitigation based on recommendations derived from AHRIC outcomes

Such a correction is, of course, possible at any step, as long as the biases are evaluated in a timely manner during progress of a project. Overall the approach ensures herewith a straight and timely project process of high quality with minimal latencies, which is required for the final disposal search-process.

## 4 Discussion

The paper discusses the importance of considering human reliability in the modelling activities for localizing the final disposal for high radioactive materials in Germany. It states that human impacts in the modelling, the parametrization and the interpretation of results will play a key role to fulfil the requirement of the final disposal act and not to fail to find the location of best possible safety.

Psychological influences discussed can be classified using the terms trust and biases. Influences can stem from:

- Hierarchies, which lead to restrictions in expressions of opinions or dependence on leadership behaviour
- Groupthink, which leads to incomplete information search or lack of inclusion of alternative points of view
- Conflicting goals, which lead to social pressure or heurisms (mental shortcuts)

Securing the reliability of the modelling process requires the assessment and mitigation of human aspects in the modelling process by applying human reliability assessments on goal-oriented behaviours which is capable to assess the modelling processes. The method AHRIC allows for this by monitoring biases and trust-issues in the modelling process. The method consists of a questionnaire that can be used either as a self-monitoring tool or independent assessment of the impact of biases and trust on modelling activities.

The tool fosters a representative and diverse set of experts and/or tools, an open-minded team and review process as well as a positive safety culture and climate. Herewith open-minded discussions of pros and cons of models, self-reflection and mutual learning is enabled; features required by the final disposal act to avoid hazardous developments in the search process. Overall the suggested approach is an important element of resilience and seeks to avoid drift-into-failure scenarios (Seidel, 2024).

If the AHRIC self-assessment tool is used for the phases of modelling, the results allow statements to be made about the existence of one or several biases. For a group, for example, after the individual values are aggregated to the mean, the result could be that a methodological bias ('We've always done it that way') and the described phenomenon of group polarisation apply. Even though cognitive biases cannot be entirely avoided, their impact can be reduced when AHRIC is applied in a structured and systematic manner that fosters awareness of the underlying processes.

Overall, at the end of the development process, a model emerges that is reliable, partially reliable or unreliable in terms of human reliability issues. Based on the results and nature of Biases revealed by the assessment, particular mitigation measures can be identified to overcome safety critical outcomes of biases in a modelling sequence or planning process in general. The AHRIC methods also include recommendations for mitigating the identified biases. These are immediately presented to the users afterwards.

## 5 Author contribution

Oliver Straeter and Fabian Fritsch contributed equally to this work.

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
