# Peer review of "How to handle uncertainties in modelling due to human reliability issues for nuclear disposals"

_Safety of Nuclear Waste Disposal, 2025_

## Author Comment (AC2)

**Reply to reviewer's comments on sand-2025-3**

Dear editors,

We thank the reviewers for their constructive feedback, which has contributed to refining and enhancing the manuscript.

The reviewers' comments have been reproduced below, with our point-by-point responses highlighted in green. All modifications described in these responses have been implemented in the revised manuscript.

Best regards,

Fabian Fritsch and Oliver Straeter

**RC1: 'Comment on sand-2025-3', Anonymous Referee #1, 14 Aug 2025**

Review

**General comments:**

The paper is addressing the various types of human bias encountered in the safety case for nuclear waste repositories, it shows ways to identify them and avoid them.

In table 1, a few more modelling steps could be added that are regularly required: completeness of data survey (when is enough enough); details and type of documentation (to support the traceability of decisions); implicit assumptions (often hidden in larger models / input files / databases / code packages). It is also advised to put "result evaluation" above "result interpretation".

In addition, the last step refers to reassessment (rather than evaluation). Concerning the parameterization of models, it is dangerous (but often observed) that modellers just use the databases coupled to code packages that they have paid for without checking the origin and quality of these parameter sets. Another point with parameterisation is the often individually biased selection of process and their uncertainty to be discussed when it comes to the categories "unknown knowns" and "unknown unkowns", i.e. how to deal with missing understanding and parameters (ignorance vs. uncertainty).

Thank you for your comment on our contribution. The purpose of Table 1 is to present generic modelling steps and to use them to illustrate the concept of human reliability. In this context, the step of re-evaluation is of considerable importance. The term "result evaluation" was never used in this publication / the comment on ""result evaluation" above "result interpretation"" is therefore not relevant; result evaluation is in this paper understood as part of "result interpretation".

**Specific comments:**

In figure 2 (although taken from another activity) "competence" should also be fed from "evaluation", in many areas benchmarking (between codes, and also with respect to real field data or experimental results) are well-respected factors to generate trust. Examples are the huge international consortia behind DECOVALEX or JOSA.

Figure 2 has been adopted from the original publication. The term competence in this context means competence of the tool (model) and not of the human.

The work explains in great detail the biases linked to group structures and behaviour. However, it should also be mentioned that in many circumstance a "four-eye-principle" could on the contrary be beneficial to steps in modelling issues. This is closely connected to the role of "external peers" and review processes.

Thank you very much for this comment. The described principle constitutes a control measure, whereas our work concentrates on the (self-)detection of cognitive biases. It should be noted that "four-eye-principles" or "external peers" may also be ineffective when both actors are affected by comparable biases.

Line 159: URS should be spelled out and a link to the project given. →done

Lines 195ff: An example would be very beneficial for the reader to understand what the entries in Figure 5 (which, by the way, could easily be turned into a table) are really meaning; currently this is very generic, data (response scale) interpretation without associated statements is not clear. Figure 6 does obviously focus already on the next step; it is not explained where the p (Success) values are coming form – and why there are only four distinct numbers in addition to 1 and 0. In addition, the computation of the numbers in the right-most column is unclear.

We have added an example illustrating the evaluation of the AHRIC and provided an explanation of the p-values and the results column.

Line 250: What is the meaning of "heurism" in that context?

We have added the explanatory term "mental shortcut" at this point.

**Technical comments:**

Lines 223-226 are strongly redundant to lines 210+. Should be merged.

We do not consider this to be strongly redundant. Section 3.2 is self-contained, focusing on the application of AHRIC in modelling.

Line 244: "and" instead of "ans" → done

An acknowledgement of TRANSENS is missing.

The AHRIC method was not developed within the context of TRANSENS. Section 2.2 refers to a study conducted as part of the TRANSENS project and is cited accordingly.

**RC2: 'Comment on sand-2025-3', Anonymous Referee #2, 07 Oct 2025**

The manuscript addresses the important subject of human reliability aspects in the initial planning and decision stages of selection for nuclear disposal waste. It adequately outlines the individual, collective and organisational as well as cross-organisational factors which influence human reliability in planning and operational processes. In addition, it highlights the aspect of cognitive bias and trust in the respective socio-cognitive processes in the context of site selection. The authors provide recommendations for the assessment and management of human reliability aspects in alignment with the VDI 4006 guideline, highlight the importance and tight coupling of human reliability with system safety and propose a method (AHRIC: Assessment of Human Reliability in Concept phases) for the pre-emptive management of human reliability in critical modelling processes of site selection. The conceptual-methodological part of the manuscript is complemented by the presentation of a

case study derived from the TRANSENS project that should provide insights of the suitability and usefulness of the approach for practice.

The manuscript is overall well structured and for the most comprehensible and insightful. However, there are two critical shortcomings that need to be considered and revised by the authors.

First, the manuscript claims to present the results of the deployment of AHRIC in the TRANSENS project. The manuscript provides rather conceptual information as well general procedural (i.e. deployment principles) and intended outcomes of the method than presents specific results. I missed information on the procedure of assessment, on the demographics of participants and the context of measurement. In addition, the manuscript only partly provides an overview of the results on the three postulated dimensions of the method and how these can be associated with the quality of decisions in site selection as a tangible outcome. It is further more not clear, whether the example provided in Section 3.2 and Fig. 6 is a fictional one (for the sake of demonstration) or results that correspond to the suggested empirical evaluation of the method.

Thank you for your comment on our contribution. We have changed the heading of Fig. 6 to indicate that this is a fictitious example. Section 2.3 uses one result of the TRANSENS study as example In the event-tree, with details available in (Muxlhanga et al., 2024); worth mentioning that this is the only point where this paper uses work of TRANSENS. AHRIC was used in the URS-Project and never TRANSENS yet. In addition, Section 2.3 was incorrectly numbered as 3.2 in the manuscript and has now been corrected.

Second, there is sparse information on the potential vulnerability of the method against cognitive biases of the participants themselves. As a self-reporting, self-assessment method AHRIC may be susceptible itself on the perpetuation of the same cognitive biases which it claims to assess and counteract. It is sufficient that the authors provide more elaborate information on the method administration and the usage process in order to provide adequate information of the respective available safeguarding aspects and the validity of the method. Further on, the postulated verification modality of the effectiveness of the method (lines 228-229) reads more like "wishful thinking" and not as a valid outcome attained through observation and measured with solid indicators. No information whatsoever has been provided on such effectiveness measures deployed in the TRANSENS project.

We have added a sentence to this effect in the discussion. As a method, AHRIC — in combination with an established safety culture — can help reduce cognitive biases among participants and modellers.

The effect described in lines 228-229 is the result of a pilot study and does not relate to the TRANSENS project.

In addition, there are some minor aspects that need to be amended:

- Some critical terms are not clearly described and/or rely highly on the respective sources and thus, may be inadequately understood or even misunderstood by the readers (goal-oriented vs task-oriented behaviour, 38-40) → We have modified the sentence structure to improve readability.
- The method need to be presented in more details. For instance: how many items for each bias/categories in the AHRIC-M/AHRIC-S; availability of statistical quality criteria (e.g. reliability, validity)

   We have included a few minor additions for clarification.
- Some abbreviations appear without previous complete description (line 159; line 169)

- → An explanation of the abbreviation has been added, and the term "THMC" has been removed, since the application of AHRIC is not confined to Thermo-Hydro-Mechanical-Chemical models.
- Occasionally the terms used appear to be vague/not clear indicating spelling mistakes (line 133) → The passage has been refined to enhance clarity and precision.
- Occasionally language use is more colloquial than appropriate in a scientific paper (line 54: "These people") → changed
- Some rather strong statements would need to be supported by respective citation (if applicable) (line 55-56) → We have added a citation for this point.
- The term "cognitive biases" is a rather well documented and established term. Seminal researchers on the subject e.g. Tverksy and Kahneman should be cited in the first generic use of the term (line 67) → done
- In general, an additional proofreading is highly recommended as some sentences are slightly incomprehensible something that is not beneficial for the manuscript.
  - → We appreciate this helpful comment and have rechecked the text to ensure the language reads as clearly as possible.

---

## Author Response (AR2)

We have coccected the mentioned Refenrence and proofread again.

The section Author contribution was not missing; it was entiteld Author contributions